# COGNITIVELY INSPIRED LEARNING OF INCREMENTAL DRIFTING CONCEPTS

## ABSTRACT

Humans continually expand their learned knowledge to new domains and learn new concepts without any interference with past learned experiences. In contrast, machine learning models perform poorly in a continual learning setting, where input data distribution changes over time. Inspired by the nervous system learning mechanisms, we develop a computational model that enables a deep neural network to learn new concepts and expand its learned knowledge to new domains incrementally in a continual learning setting. We rely on the *Parallel Distributed Processing* theory to encode abstract concepts in an embedding space in terms of a multimodal distribution. This embedding space is modeled by internal data representations in a hidden network layer. We also leverage the *Complementary Learning Systems theory* to equip the model with a memory mechanism to overcome catastrophic forgetting through implementing pseudo-rehearsal. Our model can generate pseudo-data points for experience replay and accumulate new experiences to past learned experiences without causing cross-task interference.

## 1 INTRODUCTION

Humans continually abstract *concept classes* from their input sensory data to build semantic descriptions, and then update and expand these concepts as more experiences are accumulated Widmer & Kubat (1996), and use them to express their ideas and communicate with each other Gennari et al. (1989); Lake et al. (2015). For example, "cat" and "dog" are one of the first concept classes that many children learn to identify. Most humans expand these concepts as *concept drift* occurs, e.g., incorporating many atypical dog breeds into the "dog" concept, and also incrementally learn new concept classes, e.g. "horse" and "sheep," as they acquire more experiences. Although this concept learning procedure occurs continually in humans, continual and incremental learning of concept classes remains a major challenge in artificial intelligence (AI). AI models are usually trained on a fixed number of classes and the data distribution is assumed to be stationary during model execution. Hence, when an AI model is trained or updated on sequentially observed tasks with diverse distributions or is trained on new classes, it tends to forget what has been learned before due to cross-task interference, known as the phenomenon of *catastrophic forgetting* in the literature French (1991).

Inspired by the Parallel Distributed Processing (PDP) paradigm McClelland et al. (1986); McClelland & Rogers (2003), our goal is to enable a deep neural network to learn *drifting concept classes* Gama et al. (2014) incrementally and continually in a sequential learning setting. PDP hypothesizes that abstract concepts are encoded in higher layers of the nervous system McClelland & Rogers (2003); Saxe et al. (2019). Similarly, and based on behavioral similarities between artificial deep neural networks and the nervous system Morgenstern et al. (2014) , we can assume that the data representations in hidden layers of a deep network encode semantic concepts with different levels of abstractions. We model these representations as an embedding space in which semantic similarities between input data points are encoded in terms of geometric distances Jiang & Conrath (1997), i.e., data points that belong to the same concept class are mapped into separable clusters in the embedding space. When a new concept is abstracted, a new distinct cluster should be formed in the embedding space to encode that new class. Incremental concepts learning is feasible by tracking and remembering the representation clusters that are formed in the embedding space and by considering their dynamics as more experiences are accumulated in new unexplored domains.

We benefit from the Complementary Learning Systems (CLS) theory McClelland et al. (1995) to mitigate catastrophic forgetting. CLS is based on empirical evidences that suggest experience replay of recently observed patterns during sleeping and waking periods in the human brain helps to accumulate the new experiences to the past learned experiences without causing interference McClelland et al. (1995); Robins (1995). According to this theory, hippocampus plays the role of a short-term memory buffer that stores samples of recent experiences and catastrophic forgetting is prevented by replaying samples from the hippocampal storage to implement pseudo-rehearsal in the neocortex during sleeping periods through enhancing past learned knowledge. Unlike AI memory buffers that store raw input data point, e.g., samples of raw images, hippocampal storage can only store encoded representations after some level of abstraction which suggests a generative nature.

Inspired by the above two theories, we expand a base deep neural classifier with a decoder network, which is amended from a hidden layer, to form an autoencoder with the hidden layer as its bottleneck. The bottleneck is used to model the discriminative embedding space. As a result of supervised learning, the embedding space becomes discriminative, i.e. a data cluster is formed for each concept class in the embedding space McClelland & Rogers (2003). These clusters can be considered analogous to neocortical representations in the brain, where the learned abstract concepts are encoded McClelland et al. (1986). We use a multi-modal distribution to estimate this distribution. We update this parametric distribution to accumulate new experiences to past learned experiences consistently. Since our model is generative, we can implement the offline memory replay process in the sleeping brain to prevent catastrophic forgetting McClelland et al. (1995); Rasch & Born (2013). When a new task arrives, we draw random samples from the multi-modal distribution and feed them into the decoder network to generate representative pseudo-data points. These pseudo-data points are then used to implement pseudo-rehearsal for experience replay Robins (1995). We demonstrate that the neural network can learn drifting conceptsincrementally while mitigating forgetting.

## 2 RELATED WORK

The problem of continual learning of incremental drifting concepts lies in the intersection of *lifelong learning* to encode drifting concept classes and *incremental learning* to incorporate new concept classes. In a lifelong learning setting, the number of classes are usually assumed to be fixed, but the distribution of sequential tasks is non-stationary. In an incremental learning setting, concept classes are learned sequentially while the conditional distribution for each concept class is stationary.

**Continual learning:** the major challenge of continual learning is tackling catastrophic forgetting. Previous works in the literature mainly rely on experience replay Li & Hoiem (2018). The core idea of experience replay is to implement pseudo-rehearsal by replaying representative samples of past tasks along with the current task data to retain the learned distributions. Since storing these samples requires a memory buffer, the challenge is selecting the representative samples to meet the buffer size limit. For example, selecting uncommon samples that led to maximum effect in past experiences has been found to be effective Schaul et al. (2016). However, as more tasks are learned, selecting the effective samples becomes more complex. The alternative approach is to use generative models that behave more similar to humans French (1999). Shin et al. (Shin et al. (2017)) use a generative adversarial structure to mix the distributions of all tasks. It is also feasible to couple the distributions of all tasks in the bottleneck of an autoencoder. The shared distribution then can be used to generate pseudo-samples Rannen et al. (2017).Weight consolidation using structural plasticity Lamprecht & LeDoux (2004); Zenke et al. (2017); Kirkpatrick et al. (2017) is another approach to approximate experience replay. The idea is to identify important weights that retain knowledge about a task and then consolidate them according to their relative importance for past tasks in the future.

**Incremental learning::** forgetting in *incremental learning*stems from updating the model when new classes are incorporated, rather concept drifts in a fixed number of learned classes. Hence, the goal is to learn new classes such that knowledge about the past learned classes is not overwritten. A simple approach is to expand the base network as new classes are observed. Tree-CNN Roy et al. (2020) proposes a hierarchical structure that grows like a tree when new classes are observed. The idea is to group new classes into feature-driven super-classes and find the exact label by limiting the search space. As the network grows, the new data can be used to train the expanded network. Sarwar et al. Sarwar et al. (2019) add new convolutional filters in all layers to learn the new classes through new parameters. The alternative approach is to retain the knowledge about old classes in

an embedding feature space. Rebuffi et al. Rebuffi et al. (2017) proposed iCarl which maps images into a feature space that remains discriminative as more classes are learned incrementally. A fixed memory buffer is used to store exemplar images for each observed class. Each time a new class is observed, these images are used to learn a class-level representative vector in the feature space such that the testing images can be classified using nearest neighbor with respect to these vectors.

**Contributions:** We develop a unified framework that addresses challenges of both incremental learning and lifelong learning for the first time. Our idea is based on tracking and consolidating the multimodal distribution that is formed by the internal data representations of sequential tasks in a base neural network model hidden layers. We model this distribution as a Gaussian mixture model (GMM) with time-dependent number of components. Concept drifts are learned by updating the corresponding GMM component for a particular class and new concepts are learned by adding new GMM components. We also make the model generative to implement experience replay. We provide both theoretical and experimental results to justify why our algorithm is effective.

## 3 PROBLEM STATEMENT

Consider a learning agent which observes a sequence of observed tasks $\{\mathcal{Z}^{(t)}\}_{t=1}^T$ Chen & Liu (2016) and after learning each task moves forward to learn the next task. Each task is a classification problem in a particular domain and each class represents a concept. The classes for each task can be new unobserved classes, i.e., necessitating incremental learning Rebuffi et al. (2017), or drifted forms of the past learned classes, i.e., necessitating lifelong learning Chen & Liu (2016), or potentially a mixture of both cases. Formally, a task is characterized by a dataset $\mathcal{D}^{(t)} = \langle \boldsymbol{X}^{(t)}, \boldsymbol{Y}^{(t)} \rangle$, where $\boldsymbol{X}^{(t)} = [\boldsymbol{x}_1^t, \dots, \boldsymbol{x}_n^t] \in \mathbb{R}^{d \times n_t}$ and $\boldsymbol{Y}^{(t)} \in \mathbb{R}^{k_t \times n_t}$ are the data points and one-hot labels, respectively. The goal is to train a time-dependent classifier function $f^{(t)}(\cdot) : \mathbb{R}^d \to \subset \mathbb{R}^{k_t}$- where $k_t$ is the number of classes for the $t$-th task and is fixed for each task- such that the classifier continually generalizes on the tasks seen so far. The data points $\boldsymbol{x}_i^{(t)} \sim q^{(t)}(\boldsymbol{x})$ are assumed to be drawn i.i.d. from an unknown task distribution $q^{(t)}(\boldsymbol{x})$. Figure 1 visualizes a high-level block-diagram of this continual and dynamic learning procedure. The agent needs to expand its knowledge about all the observed concepts such that it can perform well on all the previous learned domains.

Learning each task in isolation is a standard supervised learning problem. After selecting a suitable parameterized family of functions $f_\theta^{(t)} : \mathbb{R}^d \to \mathbb{R}^{k_t}$ with learnable parameters $\theta$, e.g. a deep neural network with learnable weight paramters $\theta$, we can solve for the optimal parameters using the empirical risk minimization (ERM): $\hat{\theta}^{(t)} = \arg\min_\theta \hat{e}_\theta^{(t)} = \arg\min_\theta \sum_i \mathcal{L}_d(f_\theta^{(t)}(\boldsymbol{x}_i^{(t)}), \boldsymbol{y}_i^{(t)})$, where $\mathcal{L}_d(\cdot)$ is a proper loss function. If $n_t$ is large enough, the empirical risk expectation would be a good approximation of the real expected risk function $e^{(t)}(\theta) = \mathbb{E}_{\boldsymbol{x} \sim q^{(t)}(\boldsymbol{x})}(\mathcal{L}_d(f_{\theta^{(t)}}(\boldsymbol{x}), f(\boldsymbol{x})))$. As a result, if the base parametric family is rich and complex enough for learning the task function, then the ERM optimal model generalizes well on unseen test samples that are drawn from $q^{(t)}(\boldsymbol{x})$.

For the rest of the paper, we consider the base model $f_{\theta^{(t)}}$ to be a deep neural network with an increasing output size to encode incrementally observed classes. As stated, we rely on the PDP paradigm. Hence, we decompose the deep network into an encoder sub-network $\phi_{\boldsymbol{v}}(\cdot) : \mathbb{R}^d \to \mathcal{Z} \subset \mathbb{R}^f$ with learnable parameter $\boldsymbol{v}$, e.g., convolutional layers of a CNN, and a classifier sub-network $h_{\boldsymbol{w}}(\cdot)^{k_t} : \mathbb{R}^f \to \mathbb{R}^{k_t}$ with learnable parameters $\boldsymbol{w}$, e.g., fully connected layers of a CNN, where $\mathcal{Z}$ denotes the embedding space in which the concepts will be be formed as separable clusters.

The concepts for each task are known a priori and hence new nodes are added to the classifier subnetwork output to incorporate the new classes at time $t$. We use a softmax layer as the last layer of the classifier subnetwork. Hence, we can consider the classifier to be a a maximum *a posteriori* (MAP) estimator after training. This means that the encoder network transforms the input data distribution into an internal multi-modal distribution with $k_t$ modes in the embedding space because the embedding space $\mathcal{Z}$ should be concept-discriminative for good generalization. Each concept class is represented by a single mode of this distribution. We use a Gaussian mixture model (GMM) to model and approximate this distribution (see Figure 1, middle panel). Catastrophic forgetting is the result of changes in this internal distribution when changes in the input distribution leads to updating the internal distribution heuristically. Our idea is to track changes in the data distribution and update and consolidate the internal distribution such that the acquired knowledge from past experiences is not overwritten when new experiences are encountered and learned in the future.

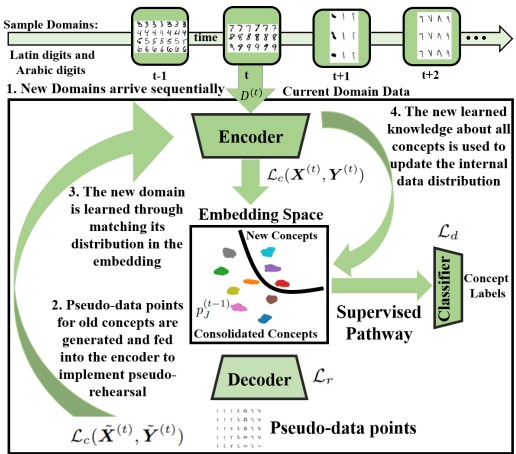

**Incremental Concept Learning Process**

Figure 1: Block-diagram visualization of the proposed Incremental Learning System including the learning procedure steps. (Based viewed enlarged on screen and in color. Enlarged version is included in the Appendix)

The main challenge is to adapt the network $f_\theta^{(t)}(\cdot)$ and the standard ERM training loss such that we can track the internal distribution continually and accumulate the new acquired knowledge consistently to the past learned knowledge with minimum interference. For this purpose, we form a generative model by amending the base model with a decoder $\psi_{\boldsymbol{u}}$ : $\mathcal{Z} \rightarrow \mathbb{R}^d$, with learnable parameters $\boldsymbol{u}$. This decoder maps back the internal representations to reconstruct the input data point in the input space such that the pair $(\phi_{\boldsymbol{u}}, \psi_{\boldsymbol{u}})$ forms an autoencoder. According to our previous discussion, a multi-modal distribution would be formed in the bottleneck of the autoencoder upon learning each task. This distribution encodes the learned knowledge about the concepts that have been learned from past experiences so far. If we approximate this distribution with a GMM, we can generate pseudo-data points that represent the previously learned concepts and use them for pseudo-rehearsal. For this purpose, we can simply draw samples from all modes of the GMM and feed these samples into the decoder subnetwork to generate a pseudo-dataset (see Figure 1). After learning each task, we can update the GMM estimate such that the new knowledge acquired is accumulated to the past gained knowledge consistently to avoid interference. By doing this procedure continually, our model is able to learn drifting concepts incrementally. Figure 1 visualizes this repetitive procedure in this lifelong learning setting.

## 4 PROPOSED ALGORITHM

When the first task is learned, there is no prior experience and hence learning reduces the following:

$$\min_{\boldsymbol{v},\boldsymbol{w},\boldsymbol{u}} \mathcal{L}_c(\boldsymbol{X}^{(1)}, \boldsymbol{Y}^{(1)}) = \min_{\boldsymbol{v},\boldsymbol{w},\boldsymbol{u}} \frac{1}{n_1} \sum_{i=1}^{n_1} \left( \mathcal{L}_d\Big(h_{\boldsymbol{w}}(\phi_{\boldsymbol{v}}(\boldsymbol{x}_i^{(1)})), \boldsymbol{y}_i^{(1)}\Big) + \gamma \mathcal{L}_r\Big(\psi_{\boldsymbol{u}}(\phi_{\boldsymbol{v}}(\boldsymbol{x}_i^{(1)})), \boldsymbol{x}_i^{(1)}\Big) \right),$$
(1)

where $\mathcal{L}_d$ is the discrimination loss, e.g., cross-entropy loss, $\mathcal{L}_r$ is the reconstruction loss for the autoencoder, e.g., $\ell_2$-norm, $\mathcal{L}_c$ is the combined loss, and $\gamma$ is a trade-off parameter between the two loss terms. When the first task is learned, also any future task, according to the PDP hypothesis, a multi-modal distribution $p^{(1)}(\boldsymbol{z}) = \sum_{j=1}^{k_1} \alpha_j \mathcal{N}(\boldsymbol{Z}|\mu_j, \Sigma_j)$ with $k_1$ components is formed in the embedding space. We assume that this distribution can be modeled with a GMM. Since the labels for the input task data samples are known, we use MAP estimation to recover the GMM parameters (see Appendix for details of this process). Let $\hat{p}^{(1)}(\boldsymbol{z})$ denotes the estimated distribution.

As subsequent tasks are learned, the internal distribution should be updated continually to accumulate the new acquired knowledge. Let $k_t = k_{old}^t + k_{new}^t$, where $k_{old}^t$ denotes the number of the previously learned concepts that exist in the current task and $k_{new}^t$ denotes the number of the new observed classes. Hence, the total number of learned concepts until $t = T$ is $k_{Tot}^T = \sum_{t=1}^T k_{new}^t$. Also, let the index set $\mathbb{N}_{Tot}^T = \{1, \dots, k_{Tot}^T\}$ denotes an order on the classes $C_j$, with $j \in \mathbb{N}_{Tot}^T$, that are observed until $t = T$. Let $\mathbb{N}_T = \mathbb{N}_{old}^T \cup \mathbb{N}_{new}^T = \{i_1, \dots, i_{k_T}\} \subset \mathbb{N}_{Tot}^T$ contains the $k_T$ indices of the existing concepts in $\mathcal{Z}^{(T)}$. To update the internal distribution after learning $\mathcal{Z}^{(T)}$, the number of distribution modes should be updated to $k_{Tot}^T$. Additionally, catastrophic forgetting must be mitigated using experience replay. We can draw random samples from the GMM distribution $\boldsymbol{z}_i \sim \hat{p}^{(T-1)}(\boldsymbol{z})$ and then pass each sample through the decoder $\psi(\boldsymbol{z}_i)$ to generate pseudo-data points for pseudo-rehearsal. Since each particular concept is represented by exactly one mode of the internal GMM distribution, the corresponding pseudo-labels for the generated pseudo-data points are known. Moreover, the confidence levels for these labels are also known from the classifier softmax

layer. To generate a clean pseudo-dataset, we can set a threshold $\tau$ and only pick the pseudo-data points for which the model confidence level is more than $\tau$. We also generate a balanced pseudo-dataset with respect to the learned classes. Doing so, we ensure suitability of a GMM with $k_{Tot}^T$ components to estimate the empirical distribution accurately after learning the subsequent tasks.

Let $\tilde{\mathcal{D}}^{(t)} = \langle \psi(\tilde{\boldsymbol{Z}}^{(t)}), \tilde{\boldsymbol{Y}}^{(t)} \rangle$ denotes the pseudo-dataset, generated at time $t$ after learning the tasks $\{\mathcal{Z}^{(s)}\}_{s=1}^{t-1}$. We form the following objective to learn the task $\mathcal{Z}^{(t)}, \forall t \geq 2$:

$$\min_{\boldsymbol{v},\boldsymbol{w},\boldsymbol{u}} \mathcal{L}_c(\boldsymbol{X}^{(t)}, \boldsymbol{Y}^{(t)}) + \mathcal{L}_c(\tilde{\boldsymbol{X}}^{(t)}, \tilde{\boldsymbol{Y}}^{(t)}) + \lambda \sum_{j \in \mathbb{N}_{old}^t} D\Big(\phi_{\boldsymbol{v}}(q^{(t)}(\boldsymbol{X}^{(t)})|C_j), \hat{p}^{(t-1)}(\tilde{\boldsymbol{Z}}^{(t)})|C_j)\Big), \quad (2)$$

where $D(\cdot, \cdot)$ is a probability distribution metric and $\lambda$ is a trade-off parameter.

---

**Algorithm 1** ICLA $(\lambda, \gamma, \tau)$

---

1: **Input:** labeled training datasets in a sequence
2: $\qquad \mathcal{D}^{(t)} = (\{\boldsymbol{X}^{(t)}, \boldsymbol{X}^{(t)}\})$ for $t =\geq 1$
3: **Initial Learning**: learn the first task via Eq. equation 1
4: **Fitting GMM:**
5: $\qquad$ estimate $\hat{p}_J^{(1)}(\cdot)$ using $\{\phi_{\boldsymbol{v}}(\boldsymbol{x}_i^{(1)}), \boldsymbol{y}_i^{(1)}\}_{i=1}^{n_t}$
6: **For** $t \geq 2$
7: $\qquad$ **Generate the pseudo dataset:**
8: $\qquad\qquad \tilde{\mathcal{D}}^{(t)} = \{(\tilde{\boldsymbol{x}}_i^{(t)} = \psi(\tilde{\boldsymbol{z}}_i^{(t)}), \tilde{\boldsymbol{y}}_i^{(t)})\}$
9: $\qquad\qquad (\tilde{\boldsymbol{z}}_i^{(t)}, \tilde{\boldsymbol{y}}_i^{(t)}) \sim \hat{p}^{(t-1)}(\cdot)$
10: $\qquad$ **Task learning:**
11: $\qquad\qquad$ learnable parameters are updated via Eq. equation 2
12: $\qquad$ **Estimating the internal distribution:**
13: $\qquad\qquad$ update $\hat{p}^{(t)}(\cdot)$ with $k_{Tot}^{(t)}$ components via the
14: $\qquad\qquad$ combined samples $\{\phi_{\boldsymbol{v}}(\boldsymbol{x}_i^{(t)}), \phi_{\boldsymbol{v}}(\tilde{\boldsymbol{x}}_i^{(t)})\}_{i=1}^{n_t}$
15: **EndFor**

---

The first and the second terms in Eq. equation 2 are combined loss terms for the current task training dataset and the generated pseudo-dataset that represent the past tasks, defined similar to Eq. equation 1. The second term in Eq. equation 2 mitigates catastrophic forgetting through pseudo-rehearsal process. The third term is a crucial term to guarantee that our method will work in a lifelong learning setting. This term enforces that each concept is encoded in one mode of the internal distribution across all tasks. This term is computed on the subset of the concept classes that are shared between the current task and the pseudo-dataset, i.e, $\mathbb{N}_{old}^t$, to enforce consistent knowledge accumulation. Minimizing the probability metric $D(\cdot, \cdot)$ enforces that the internal conditional distribution for the current task $\phi_{\boldsymbol{v}}(q^{(t)}(\cdot|C_j))$, conditioned on a particular shared concept $C_j$, to be close to the conditional shared distribution $p^{(t-1)}(\cdot|C_j)$. Hence, both form a single mode of the internal distribution and concept drifting is mitigated. Conditional matching of the two distributions is feasible as we have access to pseudo-labels. Adding this term guarantees that we can continually use a GMM with exactly $k_{Tot}^{(t)}$ components to capture the internal distribution in this lifelong learning setting. The remaining task is to select a suitable probability distance metric $D(\cdot, \cdot)$ for solving Eq. equation 2. Wasserstein Distance (WD) metric has been found to be an effective choice for deep learning due to its applicability for gradient-based optimization Courty et al. (2017). To reduce the computational burden of computing WD, we use the Sliced Wasserstein Distance (SWD) Bonneel et al. (2015). (for details on the SWD loss, please refer to the Appendix). Our Incremental Concept Learning Algorithm (ICLA) is summarized in Algorithm 1.

## 5 THEORETICAL ANALYSIS

We demonstrate that ICLA minimizes an upperbound for the expected risk of the learned concept classes across all the previous tasks for all $t$. We perform our analysis in the embedding space as an input space and consider the hypothesis class $\mathcal{H} = \{h_{\boldsymbol{w}}(\cdot)|h_{\boldsymbol{w}}(\cdot) : \mathcal{Z} \to \mathbb{R}_t^k, \boldsymbol{w} \in \mathbb{R}^H\}$. Let $e_t(\boldsymbol{w})$ denote the real risk for a given function $h_{\boldsymbol{w}^{(t)}}(\cdot) \in \mathcal{H}$ when used on task $\mathcal{Z}^{(t)}$ data representations in the embedding space. Similarly, $\tilde{e}_t(\boldsymbol{w})$ denotes the observed risk for the function $h_{\boldsymbol{w}^{(t)}}(\cdot)$ when used on the pseudo-task, generated by sampling the learned GMM distribution $\hat{p}^{(t-1)}$. Finally, let $e_{t,s}(\boldsymbol{w})$ denote the risk of the model $h_{bmw^{(t)}}(\cdot)$ when used only on the concept classes in the set $\mathbb{N}_s \subset \mathbb{N}_{Tot}^t$, for $\forall s \leq t$, i.e., task specific classes, after learning the task $\mathcal{Z}^{(t)}$.

**Theorem 1** : Consider two tasks $\mathcal{Z}^{(t)}$ and $\mathcal{Z}^{(s)}$ in our framework, where $s \leq t$. Let $h_{\boldsymbol{w}^{(t)}}$ be an optimal classifier trained for the $\mathcal{Z}^{(t)}$ using the ICLA algorithm. Then for any $d' > d$ and $\zeta < \sqrt{2}$, there exists a constant number $N_0$ depending on $d'$ such that for any $\xi > 0$ and $\min(\tilde{n}_{t|\mathbb{N}_s}, n_s) \geq$

$\max(\xi^{-(d'+2),1})$ with probability at least $1 - \xi$ for $h_{\boldsymbol{w}^{(t)}} \in \mathcal{H}$, the following holds:

$$e_s(\boldsymbol{w}) \leq e_{t-1,s}(\boldsymbol{w}) + W(\hat{p}_s^{(t-1)}, \phi(\hat{q}^{(s)})) + e_{\mathcal{C}}(\boldsymbol{w}^*) + \sqrt{(2\log(\frac{1}{\xi})/\zeta)}\left(\sqrt{\frac{1}{\tilde{n}_{t|\mathbb{N}_s}}} + \sqrt{\frac{1}{n_s}}\right), \quad (3)$$

where $W(\cdot, \cdot)$ denotes the WD metric, $\tilde{n}_{t|\mathbb{N}_s}$ denotes the pseudo-task samples that belong to the classes in $\mathbb{N}_s$, $\phi(\hat{q}^{(s)}(\cdot))$ denotes the empirical marginal distribution for $\mathcal{Z}^{(s)}$ in the embedding, $\hat{p}_s^{(t-1)}$ is the conditional empirical shared distribution when the distribution $\hat{p}^{(t-1)}(\cdot)$ is conditioned to the classes in $\mathbb{N}_s$, and $e_{\mathcal{C}}(\boldsymbol{w}^*)$ denotes the optimal model learned for the combined risk of the tasks on the shared classes in $\mathbb{N}_s$, i.e., $\boldsymbol{w}^* = \arg\min_{\boldsymbol{w}} e_{\mathcal{C}}(\theta) = \arg\min_{\boldsymbol{w}}\{e_{t,s}(\boldsymbol{w}) + e_s(\boldsymbol{w})\}$. This is a model with the best performance if the tasks could be learned simultaneously.

***Proof***: included in the Appendix due to page limit.

We then use Theorem 1 to conclude the following lemma:

**Lemma 1** : Consider the ICLA algorithm after learning $\mathcal{Z}^{(T)}$. Then all tasks $t < T$ and under the conditions of Theorem 1, we can conclude the following inequality:

$$e_t(\boldsymbol{w}) \leq e_{T-1,t}(\boldsymbol{w}) + W(\phi(\hat{q}^{(t)}), \hat{p}_t^{(t)}) + e_{\mathcal{C}}(\boldsymbol{w}^*) + \sum_{s=t}^{T-2} W(\hat{p}_t^{(s)}, \hat{p}_t^{(s+1)}) + \sqrt{(2\log(\frac{1}{\xi})/\zeta)}\left(\sqrt{\frac{1}{n_t}} + \sqrt{\frac{1}{\tilde{n}_{t|\mathbb{N}_t}}}\right),$$
$$(4)$$

***Proof***: included in the Appendix due to page limit.

Lemma 1 concludes that when a new task is learned at time $t = T$, ICLA updates the model parameters conditioned on minimizing the upper bound of $e_t$ for all $t < T$ in Eq. 4. The last term in Eq. 4 is a small constant term when the number of training data points is large. If the network is complex enough so that the PDP hypothesis holds, then the classes would be separable in the embedding space and in the presence of enough labeled samples, the terms $e_{T-1,t}(\boldsymbol{w})$ would be small because $e_{T-1}(\boldsymbol{w})$ is minimized using ERM. The term $W(\phi(\hat{q}^{(t)}), \hat{p}_t^{(t)})$ would be small because we deliberately fit the GMM distribution $\hat{p}^{(t)}$ to the distribution $\phi(\hat{q}^{(t)})$ in the embedding space when learning the task $\mathcal{Z}^{(t)}$. Existence of this term indicates that our algorithm requires that internal distribution can be fit with a GMM distribution with high accuracy and this limits applicability of our algorithm. Note however, all parametric learning algorithms face this limitation. The term $e_{\mathcal{C}}(\boldsymbol{w}^*)$ is small because we continually match the distributions in the embedding space class-conditionally. Hence, if the model is trained on task $\mathcal{Z}^{(t)}$ and the pseudo-task at $t - T$, it will perform well on both tasks. Note that this is not trivial because if the wrong classes are matched across the domains in the embedding space, the term $e_{\mathcal{C}}(\boldsymbol{w}^*)$ will not be minimal. Finally, the sum term in Eq. 4 indicates the effect of experience replay. Each term in this sum is minimized at $s = t+1$ because we draw random samples from $\hat{p}_t^{(t)}$ and then train the autoencoder to enforce $\hat{p}_t^{(t)} \approx \psi(\phi(\hat{p}_t^{(t)}))$. Since all the terms in the upperbound of $e_t(\boldsymbol{w})$ in Eq. 4 are minimized when a new task is learned, catastrophic forgetting of the previous tasks will be mitigated. Another important intuition from Eq. 4 is that as more tasks are learned after learning a task, the upperbound becomes looser as more terms are accumulated in the sum which enhances forgetting. This observation accords with our intuition about forgetting as more time passes after initial learning time of a task or concept.

## 6 EXPERIMENTAL VALIDATION

To the best of knowledge, no prior method has been developed to address challenges of both continual and incremental learning setting at the same time. For this reason, we validate our method on two sequential task learning settings: incremental learning and continual incremental learning. Incremental learning is a special case of our learning setting when each concept class is observed only in one task and concept drift does not exist. We use this special case to compare our method against existing incremental learning approaches in the literature to demonstrate that our method is comparably effective.Our implementation is available as a supplementary.

**Evaluation Methodology**: We use the same network structure for all the methods for fair comparison. To visualize the results, we generate learning curves by plotting the model performance on

the testing split of datasets versus the training epochs, i.e, to model time. We report the average performance of five runs. Visualizing learning curves allows studying temporal aspects of learning. For comparison, we provide learning curves for: (a) full experience replay (FR) which stores the whole training data for all the previous tasks and (b) experience replay using a memory buffer (MB) with a fixed size, similar to Li et. al (Li & Hoiem (2018)). At each time-step, the buffer stores an equal number of samples per concept from the previous tasks. When a new task is learned, a portion of old stored samples are discarded and replaced with samples from the new task to keep the buffer size fixed. FR serves as a best achievable upperbound to measure the effectiveness of our method against the upperbound. For details about the experimental setup, please refer to the Appendix.

## 6.1 Incremental Learning

The concept classes are encountered only at one task in incremental learning. We design two incremental learning experiments using the MNIST and the Fashion-MNIST datasets. Both datasets are classification datasets with ten classes. MNIST dataset consists of gray scale images of handwritten digits and Fashion-MNIST consists of images of common fashion products. We consider an incremental learning setting with nine tasks for the MNIST dataset. The first task is a binary classification of digits 0 and 1 and each subsequent task involves learning a new digit. The experimental setup for Fashion-MNIST dataset is similar, but we considered four tasks and each task involves learning two fashion classes. We use a memory buffer with the fixed size of 100 for MB. We build an autoencoder by expanding a VGG-based classifier by mirroring the layers.

Figure 2 presents results for the designed experiments. For simplicity, we have provided condensed results for all tasks in a single curve. Each task is learned in 100 epochs and at each epoch, the model performance is computed as the average classification rate over all the classes, observed before. We report performance on the standard testing split of each dataset for the observed classes. Figure 2a and present the learning curves for the MNIST experiments. Similarly, Figure 2b present learning curves for the Fashion-MNIST experiments. We can see in both figures that FR (dashed blue curves) leads to superior performance. This is according to expectation but as we discussed, the challenge is the requirement for a memory buffer with an unlimited size. The buffer cannot have a fixed size as the number of data points grows when more tasks are learned. MB (solid yellow curves) is initially somewhat effective and comparable with ICLA, but as more tasks are learned, forgetting effect becomes more severe. This is because fewer data points per task can be stored in the buffer with fixed size as more tasks are learned. As a result, the stored samples would not be sufficiently representative of the past learned tasks. In comparison, we can generate many pseudo-data points.

We can also see in Figure 2a and Figure 2b that ICLA (dotted green curves) is able to mitigate catastrophic forgetting considerably better than MB and the performance difference between ICLA and MB increases as more tasks are learned. We also observe that ICLA is more effective for MNIST dataset. This is because FMNIST data points are more diverse. As a result, generating pseudo-data points that look more similar to the original data points is easier for the MNIST dataset given that we are using the same network structure for both tasks. Another observation is that the major performance degradation for ICLA occurs each time the network starts to learn a new concept class as initial sudden drops. This degradation occurs due to the existing distance between the distributions $\hat{p}_{J,k}^{(T-1)}$ and $\phi(q^{(s)})$ at $t = T$ for $s < T$. Although ICLA minimizes this distance, the autoencoder is not ideal and this distance is non-zero in practice, leading to forgetting effects.

For comparison against existing works, we have listed our performance and a number of methods for incremental learning on MNIST in Table 3. Two sets of tasks for incremental learning setting have been designed using MNIST in the literature: 5 tasks (5T) setting and 2 tasks (2T) setting. In the 2T setting, two tasks are define involving digits $(0 - 4)$ and $(5 - 9)$. In the 5T setting, five binary classification tasks are defined involving digits $(0, 1)$ to $(8, 9)$. We have compared our performance against several methods, representative of prior works: CAB He & Jaeger (2018), IMM Lee et al. (2017), OWM Zeng et al. (2019), GEM Lopez-Paz & Ranzato (2017), iCarl Rebuffi et al. (2017), GSS Aljundi et al. (2019), DGR Shin et al. (2017), and MeRGAN Wu et al. (2018). The CAB, IMM, and OWM methods are based on regularizing the network weights. The GEM, iCarl, and GSS methods use a memory buffer to store selected samples. Finally, DGR and MeRGAN methods are based on generative replay similar to ICLA but use adversarial learning. We have reported the classification accuracy on the ten digit classes after learning the last task in Table 3. A memory

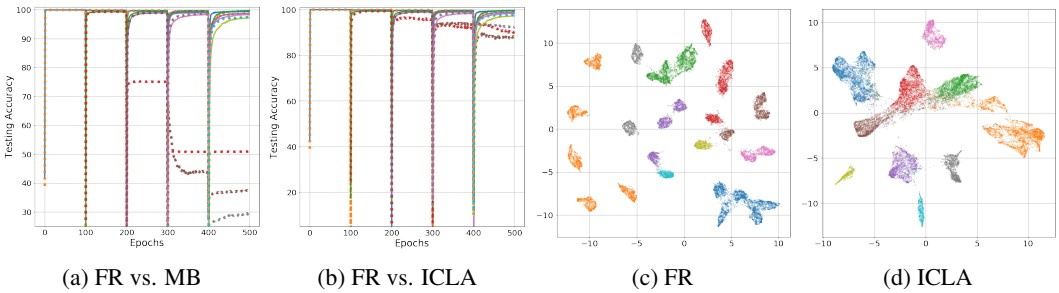

Figure 4: Learning curves for the five continual incremental learning tasks, designed using the permuted MNIST tasks (a) FR (solid) vs. MB (dotted), (b) FR (solid) vs. ICLA (dotted); UMAP visualization of (c) FR and (d) ICLA in the embedding space. (Best viewed in color on screen. See Appendix for enlarged versions.)

buffer with a fixed size of 100 is used for GEM, iCarl, and GSS. Following these works, an MLP with two layers is used as the base model for fair comparison.

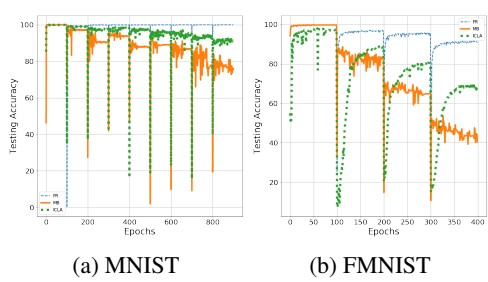

Figure 2: Learning curves for the incremental learning experiments (a) MNIST and (b) Fashion-MNIST (FMNIST) datasets; (c) MNIST performance comparison (Best viewed in color on screen. See Appendix for enlarged versions.)

We observe in Table 3 that when the buffer size is small, buffer-based methods perform poorly. Methods based on weight regularization perform quite well but note that these methods limit the network learning capacity. As a result, when the number of tasks grow, the network cannot be used to learn new tasks. Generative methods, including ICLA, perform better compared to buffer-based methods and at the same time do not limit the network learning capacity because the network weights can change after generating the pseudo-dataset. Although ICLA has the state-of-the-art performance for these tasks, there is no superior method for all conditions, because by changing the experimental setup, e.g., network structure, dataset, hyper-parameters such as memory buffer, etc, a different method may have the best performance result. However, we can conclude that ICLA has a superior performance when the network size is small and using a memory buffer is not possible.

## 6.2 CONTINUAL INCREMENTAL LEARNING

| Method | 2T | 5T |
|---|---|---|
| CAB He & Jaeger (2018) | 94.9±0.3 | - |
| IMM Lee et al. (2017) | 94.1±0.3 | - |
| OWM Zeng et al. (2019) | 96.3±0.1 | - |
| GEM Lopez-Paz & Ranzato (2017) | - | 78.0 |
| iCarl Rebuffi et al. (2017) | - | 81.0 |
| GSS Aljundi et al. (2019) | - | 61.0 |
| DGR Shin et al. (2017) | 88.7±2.6 | - |
| MeRGAN Wu et al. (2018) | 97.0 | - |
| ICLA | 97.2±0.2 | 91.6±0.4 |

Figure 3: Classification accuracy for MNIST.

Permuted MNIST task is a common supervised learning benchmark for sequential task learning Kirkpatrick et al. (2017). The sequential tasks are generated using the MNIST dataset. Each task $\mathcal{Z}^{(t)}$ is generated by rearranging the pixels of all images in the dataset using a fixed random predetermined permutation transform and keeping the labels as their original value. As a result, we can generate many tasks that are diverse, yet equally difficult. As a result, these tasks are suitable for performing controlled experiments. Since no prior work has addressed incremental learning of drifting concepts, we should design a suitable set of tasks.

We design continual incremental learning tasks that share common concepts using five permuted MNSIT tasks. The first task is a binary classification of digits 0 and 1 for the MNIST dataset. For each subsequent task, we generate a permuted MNIST task but include only the previously seen digits plus two new digits in the natural number order, e.g., the third task includes permuted versions of digit $0 - 5$. This means that at each task, new forms of all the previously learned concepts are

encountered, i.e, we need to learn drifting concepts, in additional to new tasks. Hence, the model needs to expand its knowledge about the previously learned concepts while learning new concepts. We use a memory buffer with size of 30000 for MB. Due to the nature of these tasks, we use a multi-layer perceptron (MLP) network.Figure 4 presents learning curves for the five designed permuted MNIST tasks. In this figure, the learning curve for each task is illustrated with a different color and different line styles are used to distinguish the different methods (for enlarged versions, please refer to the Appendix). At each epoch time-step, model performance is computed as the average classification rate on the standard testing split of the current and all the past learned tasks.

Figure 4a presents learning curves for MB (dotted curves) and FR (solid curves). Unsurprisingly, FR leads to almost perfect performance. We also observe MB is less effective in this setting and catastrophic forgetting is severe for MB beyond the second task. The reason is that the concepts are more diverse in these tasks. As a result, it is more challenging to estimate the input distribution using a fixed number of stored samples that also decrease due to a fixed buffer size. We can conclude that as tasks become more complex, a larger memory buffer will be necessary which poses a challenge for MB. Figure 4b presents learning curves for FR (solid curves) and MB (dotted curve). As can be seen, ICLA is able to learn drifting concepts incrementally. Again, major forgetting effect for ICLA occurs as a sudden performance drop when learning a new task starts. This observation demonstrates that an important vulnerability for ICLA is the structure of the autoencoder that we build. This can be deduced from our theoretical result because an important condition for tightness of the provided bound in Lemma 1 is that we have: $\psi \approx \phi^{-1}$. Both our theoretical and experimental results suggest that if can build auto-encoders that can generate pseudo-data points with high quality, incremental learning can be performed using ICLA. In other words, learning quality depends on the generative power the base network structure. Finally, we also observe that as more tasks are learned after learning a particular task, model performance on that particular task degrades more. This observation is compatible with the nervous system as memories fade out when time passes.

In addition to requiring a memory buffer with an unlimited size, we also demonstrate that an issue for FR is inability to identify concepts across the tasks in the embedding space. We use the UMAP McInnes et al. (2018) tool to reduce the dimensionality of the data representations in the embedding space to two for 2D data visualization. We illustrated the testing split of data for all the tasks in the embedding space $\mathcal{Z}$ in Figure 4c for FR and Figure 4d for ICLA when the final task is learned. In these figures, each color corresponds to one of the digits $\{0, 1, \ldots, 9\}$. As expected from the learning curves, data points for digits form separable clusters for both methods. This result verifies that the PDP hypothesis holds in these experiments and hence the internal distribution can be modeled using a GMM. The important distinction between FR and ICLA is that FR has led to the generation of distinct clusters for each concept class per task. This means that each concept class has not been learned internally as one concept and FR learns each concepts as several distinct concepts across the domains. This observation also serves as an ablative study for our method because it demonstrates that matching distributions class-conditionally in the embedding space is necessary for our method to work, as justified by the theoretical analysis.

In figure 4d, we observe that ten clusters for the ten observed concepts are formed when ICLA is used. This observation demonstrates that ICLA is able to track modes of the GMM successfully as more tasks are learned. ICLA is also able to build concept classes that are semantically meaningful across all tasks based on the labels. This is the reason that we can learn new classes incrementally in a continual lifelong learning scenario. In other words, as opposed to FR, ICLA encodes each cross-task concept in a single mode of the internal GMM distribution. This allows for expanding concepts for cross-domain abstraction similar to humans when new forms of concepts are observed.

## 7 CONCLUSIONS

Inspired by the CLS theory within the PDP paradigm, we developed an algorithm for continual incremental learning of concepts. Our algorithm is based on modeling the internal distribution of input data as a GMM and then updating the GMM as new experiences are acquired. We track this distribution to accumulate the new learned knowledge to the past learned knowledge consistently. We expand the base classifier model to make a generative model to allow for generating a pseudo-dataset to represent past learned tasks. The pseudo-dataset is used for pseudo-rehearsal to implement experience replay. We provided theoretical and empirical result on three benchmark datasets to validate our algorithm. Future works includes extension to domains with unlabeled data.

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

## A  APPENDIX

### A.1  COGNITIVE MODELING BACKGROUND

Our work is inspired by the "complementary learning systems" (CLS) theory within the "parallel distributed processing' (PDP) paradigm.

### A.2  PARALLEL DISTRIBUTED PROCESSING

Parallel distributed processing (PDP) approach in cognitive science tries to explain mental phenomena using structures similar to artificial neural networks McClelland et al. (1986) which were historically inspired by biological neurons and their parallel processing ability in low-level structures of the nervous system. Within this framework, learning process is modeled as adjusting weights in a network according to various rules such as Hebbian learning Song et al. (2000). PDP models data representations in the nervous system as distributed representations that are encoded in the neural activation functions Hinton et al. (1984) which is analogous to representing data in a semantically meaningful embedding space. Hasson et al. Hasson et al. (2020) argue that although evolution trains the biological neural networks blindly based on behavioral advantage, but the emerging behaviors are similar to behaviors that are observed in the artificial neural networks. They argue that both biological and artificial neural networks learn a meaningful embedding space by optimizing an objective function on densely sampled training data, i.e., empirical risk minimization. As a result, the dimensions of the embedding space capture features that help to encode informative variations across the input data points. We have based our work on this hypothesis. This means that when we train an artificial neural networks for classification, data representations encode input data similarity in terms of belonging to the same class. This means that we model the data representation distribution using a multi-modal distribution.

### A.3  COMPLEMENTARY LEARNING SYSTEMS

We rely on the Complementary Learning Systems (CLS) theory McClelland et al. (1995) to prevent catastrophic forgetting both when concepts drift or when new concepts are observed. CLS theory is proposed within the PDP paradigm and hypothesizes that continual lifelong learning ability of the nervous system is a result of a dual long- and short-term memory system. The hippocampus acts as short-term memory and encodes recent experiences that are used to consolidate the knowledge in the neocortex as long-term memory through offline experience replays during sleep Diekelmann & Born (2010). The hippocampal experience replay is more of a generative process because the input stimuli is absent during these replays. In our work the internal multimodal distribution models the neocortical consolidated knowledge. When a task is learned, this distribution is updated to incorporate the new learned knowledge to update the long-term memory. The hippocampal experience replay is modeled when the pseudo-dataset is generated to prevent catastrophic forgetting using pseudo-rehearsal. This pseudo-dataset is more representative of the recent memory, as demonstrated by both our theoretical and empirical results.

### A.4  BLOCK DIAGRAM OF THE PROPOSED METHOD

Figure 5 presents an enlarged version of the system block-diagram for more clarity on how the PDP and the CLS theories are reflected in our model. Visualization of the data representation in the embedding space in Figure 5 highlights the PDP hypothesis. An important condition for the proposed method to work is that the PDP hypothesis holds. This means that the concepts are formed as clusters in the embedding. As a result, the task data in the embedding would follow a GMM distribution and the number of components of this GMM is equal to the number of observed classes. As a result, the second term in Eq. (3) is the distance between the empirical and the real distributions for a GMM. Hence, the second term is minimized by fitting a GMM on the drawn distribution samples. Similar to all the parametric algorithms, our method works only if the assumption about the data distribution is correct. All parametric algorithms are limited in this sense.

After learning each task, we need to update the estimate of the internal GMM distribution in two aspects for generating representative pseudo-dataset in the future. First, the number of components should be updated to $k_t$ because new classes may be observed. Second, estimates for parameters

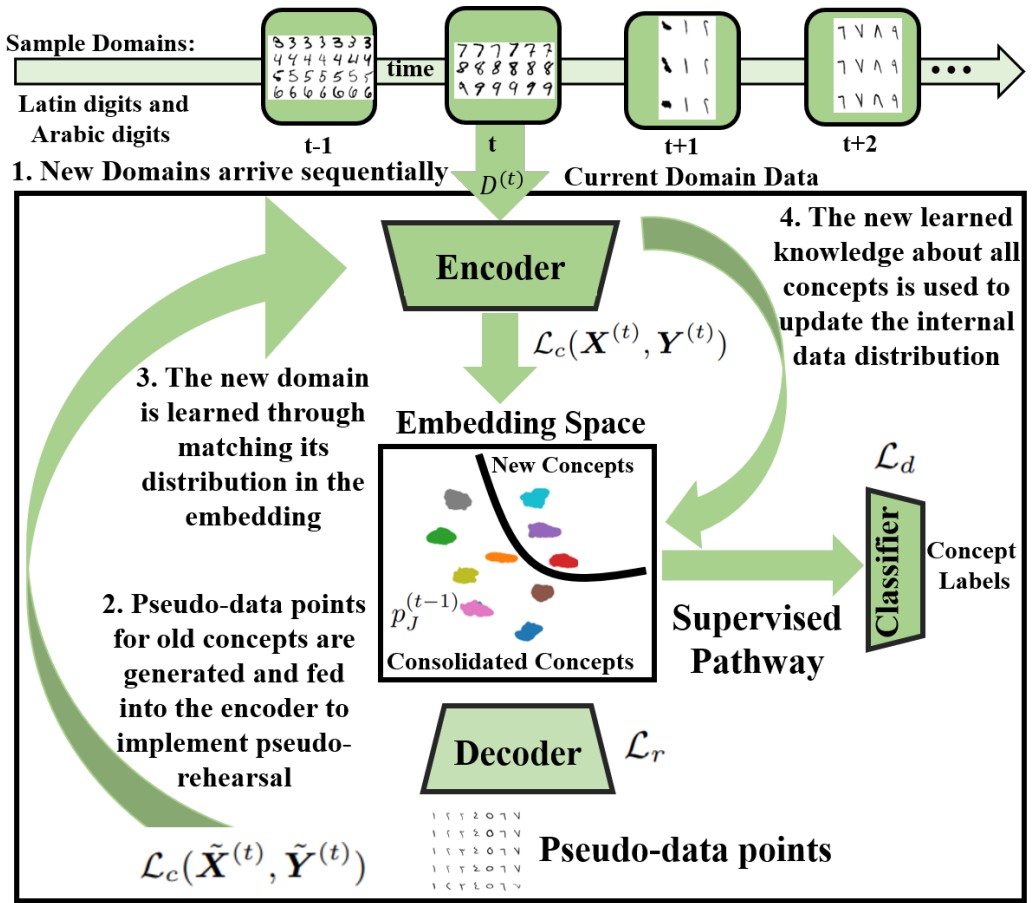

**Incremental Concept Learning Process**

Figure 5: Block-diagram Architecture of the proposed Incremental Learning System.

of each concept cluster, i.e., the mean and the variance of the corresponding Gaussian component, should be updated to incorporate potential concept drifts. Updating this distribution models the process of knowledge consolidation in the nervous system using recent experiences.

## A.5   GMM ESTIMATION

Upon learning a task, the internal distribution will be updated according to the input distribution. The empirical version of the internal distribution is encoded by the training data samples $\{(\phi_{\boldsymbol{v}}(\boldsymbol{x}_i^{(t)}), \boldsymbol{y}_i^{(t)})\}_{i=1}^{n_t}$, where with a slight abuse of notation, we use the same notation to denote the pseudo-samples. We consider the distribution $p^{(t)}(\boldsymbol{z})$ to be a GMM with $k_t$ components:

$$p^{(t)}(\boldsymbol{z}) = \sum_{j=1}^{k_t} \alpha_j \mathcal{N}(\boldsymbol{z}|\boldsymbol{\mu}_j, \boldsymbol{\Sigma}_j), \tag{5}$$

where $\alpha_j$ denotes the mixture weights, i.e., prior probability for each class, $\boldsymbol{\mu}_j$ and $\boldsymbol{\Sigma}_j$ denote the mean and co-variance for each component. Since we have labeled data points, we can compute the GMM parameters using MAP estimates. Let $\boldsymbol{S}_j$ denote the support set for class $j$ in the training dataset, i.e., $\boldsymbol{S}_j = \{(\boldsymbol{x}_i^{(t)}, \boldsymbol{y}_i^{(t)}) \in \mathcal{D}_{\mathcal{S}} | \arg\max \boldsymbol{y}_i^{(t)} = j\}$. Then, the MAP estimate for the

parameters would be:

$$
\hat{\alpha}_j = \frac{|\boldsymbol{S}_j|}{n_t},
$$

$$
\hat{\boldsymbol{\mu}}_j = \sum_{(\boldsymbol{x}_i^{(t)}, \boldsymbol{y}_i^{(t)}) \in \boldsymbol{S}_j} \frac{1}{|\boldsymbol{S}_j|} \phi_v(\boldsymbol{x}_i^{(t)}), \quad \hat{\boldsymbol{\Sigma}}_j = \sum_{(\boldsymbol{x}_i^{(t)}, \boldsymbol{y}_i^{(t)}) \in \boldsymbol{S}_j} \frac{1}{|\boldsymbol{S}_j|} \big(\phi_v(\boldsymbol{x}_i^{(t)}) - \hat{\boldsymbol{\mu}}_j\big)^\top \big(\phi_v(\boldsymbol{x}_i^{(t)}) - \hat{\boldsymbol{\mu}}_j\big). \tag{6}
$$

We can use these estimates to draw samples from $\hat{p}^{(t)}(\cdot)$ to generate a representative pseudo-dataset before learning the subsequent task.

### A.6 PROOF OF THEOREM 1 AND LEMMA 1

Our proof is modeled after Redko et al. Redko et al. (2017). The proof by Redko et al. Redko et al. (2017) is limited to the problem of domain adaptation in which the same classes exist across two domains. We adapt the proof to work in our learning setting, where the two distributions share only a subset the classes.

We first review the definition of the optimal transport. Let $\Omega \subset \mathbb{R}^d$ be a measurable space and $\mathcal{P}(\Omega)$ denote the set of probability distributions that are defined over $\Omega$. Given two distributions $p(\cdot), q(\cdot) \in \mathcal{P}(\Omega)$ and the cost function $c : \Omega^2 \to \mathbb{R}^+$, the optimal transport distance between $p(\cdot)$ and $q(\cdot)$ is defined as:

$$
W(p, q) = \inf_{\gamma \in \Pi(p,q)} \int_{\Omega^2} c(\boldsymbol{x}, \boldsymbol{y}) d\gamma(\boldsymbol{x}, \boldsymbol{y}), \tag{7}
$$

where $\Pi(\cdot, \cdot)$ denotes the set of all joint distributions over $\Omega^2$ that have marginal distributions $p$ and $q$. Optimal transport is well-defined for any proper selection of the cost function. In our proof, we consider that the cost function has the specif form: $c(\boldsymbol{x}, \boldsymbol{y}) = \|\eta(\boldsymbol{x}) - \eta(\boldsymbol{y})\|_{\mathcal{G}}$, where $\eta : \mathbb{R}^d \to \mathbb{R}^{d'}$ is an embedding function and $\| \cdot \|_{\mathcal{G}}$ denotes the norm function in this space.

We will need the following lemma in our proof.

**Lemma 2**: Consider two distribution $p, q \in \mathcal{P}(\Omega)$ and two functions $h_{\boldsymbol{w}}, h_{\boldsymbol{w}'} \in \mathcal{H}$, and the cost function $c(\boldsymbol{x}, \boldsymbol{y}) = \|\eta(\boldsymbol{x}) - \eta(\boldsymbol{y})\|_{\mathcal{G}}$. Assume that the hypothesis space $\mathcal{H}$ is a Reproducing Kernel Hilbert Space (RKHS) equipped with a kernel, induced by by the feature map $\eta : \Omega \to \mathbb{R}^{d'}$. Let the loss function $\mathcal{L}(\cdot, \cdot)$ to be a mathematical metric which is convex and bounded by 1. Additionally, we assume that the loss function para metrically depends on $\|h_{\boldsymbol{w}}(\boldsymbol{x}) - h_{\boldsymbol{w}'}(\boldsymbol{x})\|_{\mathcal{H}}$. Then the following inequality holds:

$$
\mathbb{E}_{\boldsymbol{x} \sim p}(\mathcal{L}_d(h_{\boldsymbol{w}'}(\boldsymbol{x}), h_{\boldsymbol{w}}(\boldsymbol{x}))) - \mathbb{E}_{\boldsymbol{x} \sim q}(\mathcal{L}_d(h_{\boldsymbol{w}'}(\boldsymbol{x}), h_{\boldsymbol{w}}(\boldsymbol{x}))) \leq W(p, q) \tag{8}
$$

*Proof:* First note that since the difference $h_{\boldsymbol{w}}(\boldsymbol{x}) - h_{\boldsymbol{w}'}(\boldsymbol{x})$ lies in the hypothesis space, then the loss function is nonlinear function that maps a member of the $\mathcal{H}$ to positive numbers. Using results from Saitoh (1997), we can deduce a scalar RKHS space $\mathcal{G}$ is formed. Following the above assumptions, we can deduce:

$$
\begin{aligned}
&\mathbb{E}_{\boldsymbol{x} \sim p}(\mathcal{L}_d(h_{\boldsymbol{w}'}(\boldsymbol{x}), h_{\boldsymbol{w}}(\boldsymbol{x}))) - \mathbb{E}_{\boldsymbol{x} \sim q}(\mathcal{L}_d(h_{\boldsymbol{w}'}(\boldsymbol{x}), h_{\boldsymbol{w}}(\boldsymbol{x}))) = \\
&\mathbb{E}_{\boldsymbol{x} \sim p}(\langle \mathcal{L}, \eta(\boldsymbol{x}) \rangle - \mathbb{E}_{\boldsymbol{x} \sim q}(\langle \mathcal{L}, \eta(\boldsymbol{x}) \rangle_{\mathcal{G}}) = \\
&\langle \mathbb{E}_{\boldsymbol{x} \sim p}(\eta(\boldsymbol{x})) - \mathbb{E}_{\boldsymbol{x} \sim q}(\eta(\boldsymbol{x})), \mathcal{L} \rangle_{\mathcal{G}} \leq \\
&\|\mathcal{L}\|_{\mathcal{G}} \|\mathbb{E}_{\boldsymbol{x} \sim p}(\eta(\boldsymbol{x})) - \mathbb{E}_{\boldsymbol{x} \sim q}(\eta(\boldsymbol{x}))\| = \| \int_{\Omega} \eta d(p - q)\|_{\mathcal{G}} = \\
&\int_{\Omega^2} \|(\eta(\boldsymbol{x}) - \eta(\boldsymbol{y})) d\gamma(\boldsymbol{x}, \boldsymbol{y})\|_{\mathcal{G}} \leq \\
&\int_{\Omega^2} \|\eta(\boldsymbol{x}) - \eta(\boldsymbol{y})\|_{\mathcal{G}} d\gamma(\boldsymbol{x}, \boldsymbol{y}) \leq \\
&\inf_{\gamma \in \Pi(p,q)} \int_{\Omega^2} \|\eta(\boldsymbol{x}) - \eta(\boldsymbol{y})\|_{\mathcal{G}} d\gamma(\boldsymbol{x}, \boldsymbol{y}) = W(p, q)
\end{aligned} \tag{9}
$$

In the first and the second lines, we have used the reproducing property in $\mathcal{G}$ space. In the third and fourth lines, we first used the property of the expectation and then inner-product property. In the

fifth and sixth lines, we have used the property of the joint distribution and then the definition of the optimal transport. We note that this proof is specific to a particular form of cost functions.

We also need the following result on convergence of the empirical distribution to the real distribution in the optimal transport norm in our proof.

**Theorem 2** (Theorem 1.1 Bolley et al. (2007)): consider that $p(\cdot) \in \mathcal{P}(\Omega)$ and $\int_\Omega \exp(\alpha \|\boldsymbol{x}\|_2^2) dp(\boldsymbol{x}) < \infty$ for some $\alpha > 0$. Let $\hat{p}(\boldsymbol{x}) = \frac{1}{N} \sum_i \delta(\boldsymbol{x}_i)$ denote the empirical distribution that is built from the samples $\{\boldsymbol{x}_i\}_{i=1}^N$ that are drawn i.i.d from $\boldsymbol{x}_i \sim p(\boldsymbol{x})$. Then for any $d' > d$ and $\xi < \sqrt{2}$, there exists $N_0$ such that for any $\epsilon > 0$ and $N \geq N_o \max(1, \epsilon^{-(d'+2)})$, we have:

$$P(W(p, \hat{p}) > \epsilon) \leq \exp(-\frac{-\xi}{2} N \epsilon^2) \tag{10}$$

We combine the above result and the previous lemma to prove Theorem 1.

**Theorem 1**: Consider two tasks $\mathcal{Z}^{(t)}$ and $\mathcal{Z}^{(s)}$ in our framework, where $s \leq t$. Let $h_{\boldsymbol{w}^{(t)}}$ be an optimal classifier trained for the $\mathcal{Z}^{(t)}$ using the ICLA algorithm. Then for any $d' > d$ and $\zeta < \sqrt{2}$, there exists a constant number $N_0$ depending on $d'$ such that for any $\xi > 0$ and $\min(n_{er,t|\mathbb{N}_s}, n_s) \geq \max(\xi^{-(d'+2)}, 1)$ with probability at least $1 - \xi$ for $h_{\boldsymbol{w}^{(t)}} \in \mathcal{H}$, the following holds:

$$\begin{aligned}
e_s(\boldsymbol{w}) \leq & e_{t-1,s}(\boldsymbol{w}) + W(\hat{p}_s^{(t-1)}, \phi(\hat{q}^{(s)})) + e_{\mathcal{C}}(\boldsymbol{w}^*) + \\
& \sqrt{(2 \log(\frac{1}{\xi})/\zeta)} \left( \sqrt{\frac{1}{n_{er,t|\mathbb{N}_s}}} + \sqrt{\frac{1}{n_s}} \right),
\end{aligned} \tag{11}$$

where $W(\cdot, \cdot)$ denotes the optimal transport distance, $n_{er,t|\mathbb{N}_s}$ denotes the subset of samples of the pseudo-task that belong to the classes in $\mathbb{N}_s$, $\phi(\hat{q}^{(s)}(\cdot))$ denotes the empirical marginal distribution for $\mathcal{Z}^{(s)}$ in the embedding space, $\hat{p}_s^{(t-1)}$ denotes the conditional empirical shared distribution when the distribution $\hat{p}^{(t-1)}(\cdot)$ is conditioned to the classes in $\mathbb{N}_s$, and $e_{\mathcal{C}}(\boldsymbol{w}^*)$ denotes the optimal model for the combined risk of the two tasks on the shared classes in $\mathbb{N}_s$, i.e., $\boldsymbol{w}^* = \arg\min_{\boldsymbol{w}} e_{\mathcal{C}}(\theta) = \arg\min_{\boldsymbol{w}} \{e_{t,s}(\boldsymbol{w}) + e_s(\boldsymbol{w})\}$.

*Proof:*

$$\begin{aligned}
e_s(\boldsymbol{w}) \leq & e_s(\boldsymbol{w}^*) + \mathbb{E}_{\boldsymbol{x} \sim \phi(q^{(s)})}(\mathcal{L}_d(h_{\boldsymbol{w}^*}(\boldsymbol{x}), h_{\boldsymbol{w}}(\boldsymbol{x}))) = \\
& \Big\{ e_s(\boldsymbol{w}^*) + \mathbb{E}_{\boldsymbol{x} \sim \phi(q^{(s)})}(\mathcal{L}_d(h_{\boldsymbol{w}^*}(\boldsymbol{x}), h_{\boldsymbol{w}}(\boldsymbol{x}))) + \\
& \mathbb{E}_{\boldsymbol{x} \sim \hat{p}_s^{(t-1)}}(\mathcal{L}_d(h_{\boldsymbol{w}^*}(\boldsymbol{x}), h_{\boldsymbol{w}}(\boldsymbol{x}))) \\
& - \mathbb{E}_{\boldsymbol{x} \sim \hat{p}_s^{(t-1)}}(\mathcal{L}_d(h_{\boldsymbol{w}^*}(\boldsymbol{x}), h_{\boldsymbol{w}}(\boldsymbol{x}))) \Big\} \leq \\
& e_s(\boldsymbol{w}^*) + \mathbb{E}_{\boldsymbol{x} \sim \hat{p}_s^{(t-1)}}(\mathcal{L}_d(h_{\boldsymbol{w}^*}(\boldsymbol{x}), h_{\boldsymbol{w}}(\boldsymbol{x}))) + W(p_s^{(t-1)}, \phi(q^{(s)})) \leq \\
& e_s(\boldsymbol{w}^*) + e_{t-1,s}(\boldsymbol{w}) + e_{t-1,s}(\boldsymbol{w}^*) + W(p_s^{(t-1)}, \phi(q^{(s)})) = \\
& e_{t-1,s}(\boldsymbol{w}) + e_{\mathcal{C}}(\boldsymbol{w}^*) + W(p_s^{(t-1)}, \phi(q^{(s)})) \leq \\
& e^{t-1,s}(\boldsymbol{w}) + e_{\mathcal{C}}(\boldsymbol{w}^*) + W(p_s^{(t-1)}, \hat{p}_s^{(t-1)}) + W(\hat{p}_s^{(t-1)}, \phi(q^{(s)})) \leq \\
& e_{t-1,s}(\boldsymbol{w}) + W(\hat{p}_s^{(t-1)}, \phi(\hat{q}^{(s)})) + e_{\mathcal{C}}(\boldsymbol{w}^*) + \\
& W(p_s^{(t-1)}, \hat{p}_s^{(t-1)}) + W(\phi(\hat{q}^{(s)}), \phi(q^{(s)})) \leq \\
& e_{t-1,s}(\boldsymbol{w}) + W(\hat{p}_s^{(t-1)}, \phi(\hat{q}^{(s)})) + e_{\mathcal{C}}(\boldsymbol{w}^*) \\
& + \sqrt{(2 \log(\frac{1}{\xi})/\zeta)} \left( \sqrt{\frac{1}{\tilde{n}_{t|\mathbb{N}_s}}} + \sqrt{\frac{1}{n_s}} \right)
\end{aligned} \tag{12}$$

In the above proof, fifth line is deduced from Lemma 1. In the sixth, we have used the triangular inequality on the loss function. In the seventh line, we have used the definition of the joint optimal model. In the lines eighth to tenth, we have used the triangular inequality on the optimal transport. In the last two lines, we have used Theorem 2.

We can now use Theorem 1 to deduce Lemma 1.

**Lemma 1** : Consider the ICLA algorithm after learning $\mathcal{Z}^{(T)}$. Then all tasks $t < T$ and under the conditions of Theorem 1, we can conclude the following inequality:

$$e_t(\boldsymbol{w}) \leq e_{T-1,t}(\boldsymbol{w}) + W(\phi(\hat{q}^{(t)}), \hat{p}_t^{(t)}) + e_{\mathcal{C}}(\boldsymbol{w}^*) +$$
$$\sum_{s=t}^{T-2} W(\hat{p}_t^{(s)}, \hat{p}_t^{(s+1)}) + \sqrt{\left(2\log(\frac{1}{\xi})/\zeta\right)}\left(\sqrt{\frac{1}{n_t}} + \sqrt{\frac{1}{\tilde{n}_{t|\mathbb{N}_t}}}\right), \tag{13}$$

*Proof*: We consider $\mathcal{Z}^{(t)}$ with empirical the distribution $\phi(\hat{q}^{(t)})$ in the embedding space and the pseudo-task with the distribution $\hat{p}^{(T-1)}$ in Theorem 1. Applying the triangular inequality on the term $W(\phi(\hat{q}^{(t)}), \hat{p}_t^{(T-1)})$ recursively, i.e., $W(\phi(\hat{q}^{(t)}), \hat{p}_t^{(T-1)}) \leq W(\phi(\hat{p}^{(t)}), \hat{p}_t^{(T-2)}) + W(\hat{p}_t^{(T-2)}, \hat{p}_t^{(T-1)})$ for all $t \leq s < T$ concludes Lemma 1.

## A.7 DETAILS OF EXPERIMENTAL IMPLEMENTATION

### A.7.1 DATASETS

We investigate the empirical performance of our proposed method using two commonly used benchmark datasets: MNIST ($\mathcal{M}$) and Fashion-MNIST ($\mathcal{U}$). MNIST is a collection of hand written digits in $28 \times 28$ pixels with 60000 and 10000 training and testing data points, respectively. Fashion-MNIST has similar properties but the images are more realistic. To generate permuted MNIST tasks, we followed the literature and applied a fixed random permutation to all the MNIST data points for generating each sequential task. We used cross entropy loss as the discrimination loss and the Euclidean norm as the Reconstruction loss. We used Keras for implementation and ADAM optimizer. We run our code on a cluster node equipped with 2 Nvidia Tesla P100-SXM2 GPU's.

### A.7.2 EVALUATION METHODOLOGY

All these datasets have their own standard testing splits. For each experiment, we used these testing splits to measure performance of the methods that we report in terms of classification accuracy. We used classification rate on the testing set of all the learned tasks to measure performance of the algorithms. At each training epoch, we compute the performance on the testing split of these tasks to generate the learning curves. We performed 5 learning trials on the training sets and reported the average performance on the testing sets for these trials. We used brute force search to cross-validate the parameters for each sequential task.

### A.7.3 NETWORK STRUCTURE

For visual recognition experiments, we used a convolutional structure as spatial visual similarity can be captured by convolutional structures. We used a VGG16-based encoder. The decoder subnetwork is generated by mirroring this structure. We flatten the last convolutional layer response and used a dense layer to form the embedding space with dimension 64. The classifier subnetwork is a single layer with sigmoid.

Following the literature, we have used an MLP with two layers for tasks of Table 1. The first layers has 100 nodes and the second layer has nodes equal to the number of learned concepts.

For permuted MNIST experiments, we used an MLP network. This selection is natural as the concepts are related through permutations which can be learned with an MLP structure better. For this reason, the images were normalized and converted to $784 \times 1$ vectors. The network had three hidden layers with 512, 256, and 32 nodes, respectively. We used ReLu activation between the hidden layers and selected the third hidden layer as the embedding space. This selection is natural because the last hidden layer, supposedly should respond to more abstract concepts. The decoder subnetwork is generated by mirroring the encoder subnetwork and the classifier subnetwork is a one layer with 10 nodes and sigmoid activation.

## A.8 ENLARGED FIGURES

For possibility of better inspection by readers, enlarged versions of Figure 2 and Figure 3 in the main body of the paper are provided in this section.

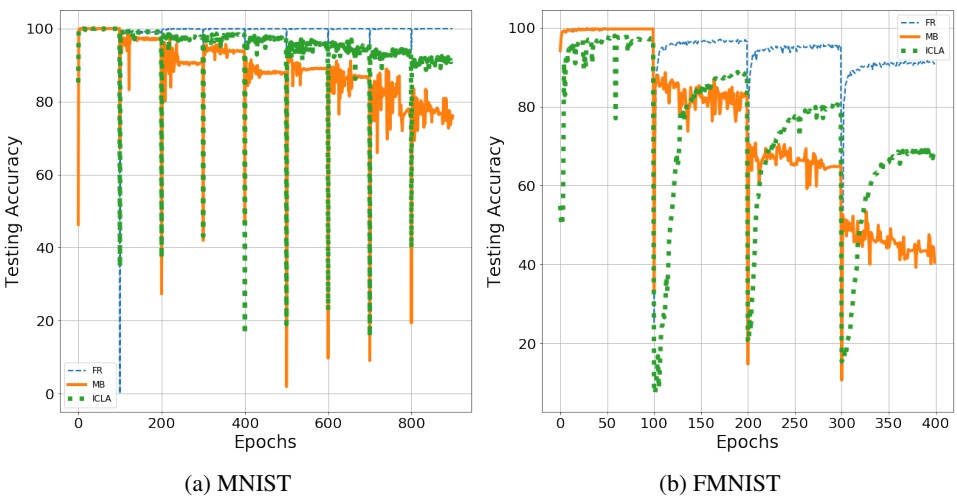

(a) MNIST                                  (b) FMNIST

Figure 6: Learning curves for the incremental learning experiments (a) MNIST and (b) Fashion-MNIST (FMNIST) datasets; (c) MNIST performance comparison (Best viewed in color on screen. Enalarged version are included in the Appendix.)

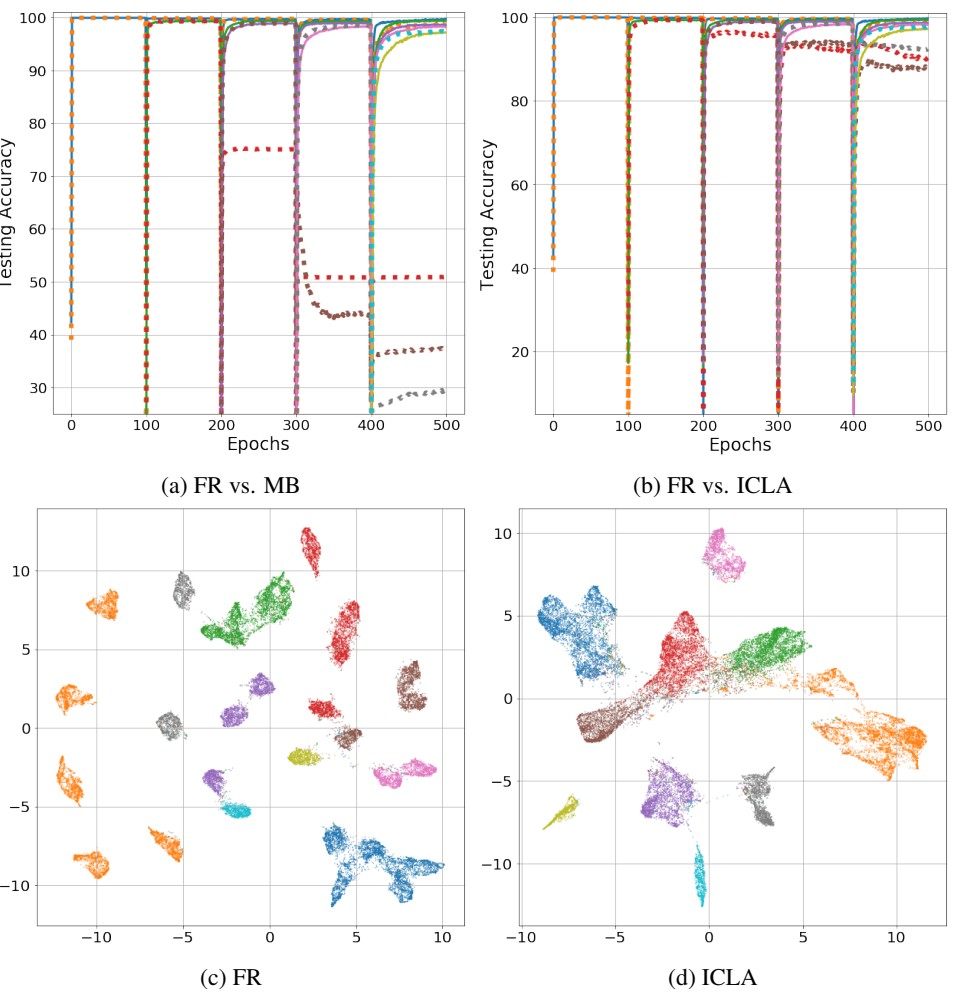

(a) FR vs. MB

(b) FR vs. ICLA

(c) FR

(d) ICLA

Figure 7: Learning curves for the five continual incremental learning tasks, designed using the permuted MNIST tasks (a) FR (solid) vs. MB (dotted), (b) FR (solid) vs. ICLA (dotted); UMAP visualization of (c) FR and (d) ICLA in the embedding space. (Best viewed in color on screen)

