# OpenReview forum: "Cognitively Inspired Learning of Incremental Drifting Concepts"
_ICLR.cc/2022/Conference — ICLR 2022 Submitted_

### Official Review · Reviewer_QevD · 2021-10-20

**Correctness:** 4
**Technical Novelty And Significance:** 3
**Empirical Novelty And Significance:** 3
**Recommendation:** 6
**Confidence:** 3

**Main Review:**

The submission is interesting and well motivated. The main novelty of the approach is fitting the latent space of the auto-encoder structure with a Gaussian mixture model which is then used to help with both generation of pseudo samples and to prevent concept drift. The final term of Eq. 2 which is used to minimize this concept drift is particularly interesting.

My main criticism of the paper is that the authors spend very little time talking about the consequences of the unimodality assumption which is the backbone of their work. To be more specific, the main innovation in this paper is the use of a GMM to interpret the latent space of an auto-encoder. This assumes that each concept is unimodal which is a reasonable assumption. However, the authors do not explore the consequences of this assumption in cases were it is not satisfied. As an example when the concepts of a new task have non-trivial relationship with previous tasks (e.g. task 1 is classifying color and task 2 is classifying shape), clearly one modality per class is no longer satisfied. What happens in such cases or in other cases where the semantic relationship between the classes is more complicated than ‘same’ or ‘new’? What are the other failure modes of the approach?

The authors spend valuable real estate to discuss PDP and CLS to argue that concepts should be encoded in a deep layer of a network and that generative processes can help alleviate catastrophic forgetting. Neither of these ideas are new. The paper can be strengthened by focusing more on what is novel and less on what has come before.

**Summary Of The Paper:**

The submission proposes a new unified methodology for continual and incremental learning based on Parallel Distributed Processing theory and Complementary Learning Systems theory. The authors prove that their methodology optimizes an upper bound for the combination of previous tasks and also demonstrate the effectiveness of their approach empirically.

**Summary Of The Review:**

The submission is a step forward in implementing approaches that can learn in a continual manner. The analytic demonstration that the loss is an upper bound is a cherry on top. However, the submission can be greatly improved if the authors focus more on the consequences of what is in fact novel in their approach (i.e. the GMM on top of the autoencoder latent), providing more experiments and a deeper discussion of the failure modes.

---

### Official Review · Reviewer_vhwe · 2021-11-02

**Correctness:** 3
**Technical Novelty And Significance:** 4
**Empirical Novelty And Significance:** 3
**Recommendation:** 5
**Confidence:** 3

**Main Review:**

Generally, I like the strong theoretical background of the paper and approach. The paper builds on important theories that are central in lifelong learning (PDP, CLS) and mathematical approaches GMM, WD. The formalism, both in the paper and in the appendix seems solid although I did not carefully analyse all the steps of the mathematical derivations.

My main concerns are two.

(1) The narrative of the paper is not easy to follow.

For example, the abstract does not mention drifting concepts that is present in the title, nor GMM that are mention in the ‘contribution’ section. However, these are central parts of where the paper aims to advance knowledge, and yet are not stated in the abstract, which makes it hard to find the claim and objectives of the paper. As a consequence, the abstract reads to generic and does not address core ideas in the paper.

One other aspect that makes it difficult to follow the paper is that the terms "lifelong learning", "incremental learning", "continual learning”. “continual incremental learning”, and “sequential learning" are not well defined and are used somehow loosely in the paper. It is conceivable that different researchers in the field might have different understanding how what each term means, and thus I think that the paper needs to explain more clearly and formally each problem. This confusion makes it difficult to  understand what problem is addressed specifically in each section. For example, differently from what appears to transpire from the paper, I assume that lifelong learning is a higher level term that include both incremental and continual learning. In other words, I would argue against the sentence "In a lifelong learning setting, the number of classes are usually assumed to be fixed”. Similarly, I’m also confused by the use of the terms ‘concept’ ‘concept class’ and ‘class’ as sometimes they are used in the same sentence but it's unclear whether they mean the same thing or not. It would be helpful if the text explains what the authors mean for each of such term.

The results in Fig 4 are difficult to read and to me not very clear (particularly in the graphs at the left).

Minor observation: Fig 2 and 3 appear after Fig.4 and all figures are difficult to read because too small.

(2) The simulation results appear limited or do not illustrate well the potential of the method.

The continual learning dynamics in Fig. 2 and 4 for the novel approach (ICLA) are compared only against standard baselines that include the full experience replay (FR) and memory buffer replay (MB). But how do these compare against well established continual learning preservation mechanisms such as EWC or MAS?

The main result for continual incremental learning in Fig 4 are not compared with the baselines that are instead shown in Fig 3. It's not clear to me why.



**Summary Of The Paper:**

This paper proposed an algorithm for learning concepts incrementally taking inspiration from Parallel Distributed Processing and Complementary Learning Systems. The core idea is to use Gaussian Mixture Models and update them incrementally plus exploit a generative model to perform pseudo-rehearsal. The paper presents a methodology and follow-up experimental results that are shown to perform well.

**Summary Of The Review:**

The paper addresses in important challenge in lifelong learning, i.e., to be able to generalise across a variety of classes that are learned sequentially through different tasks, meeting both new classes and drift, when the distribution of one class changes. The method appears sound and has a strong theoretical background. However, I feel that the paper could be improved significantly in clarity, readability, and presentation of results. It's not clear to me what are the main claims of the paper, and while some results appear to be good, the final set of results (fig 4) are difficult to interpret. In short, this paper has a strong potential, but requires a better presentation and a better illustration of the results and baselines for the last experiments in Fig. 4.

---

### Official Review · Reviewer_WRij · 2021-11-03

**Correctness:** 2
**Technical Novelty And Significance:** 1
**Empirical Novelty And Significance:** 1
**Recommendation:** 3
**Confidence:** 4

**Main Review:**

The idea of this paper is kind of reasonable but the significance and novelty are rather incremental. There are quite a few prior works that have proposed to apply VAE in continual learning and there are also some works that have proposed to generate pseudo samples for replay.  The discussion with related work is very insufficient. There are very close prior works that should be discussed and compared, such as [1,2,3,4,5,6]. Using the GMM model to simulate a multi-modal distribution is a rather naive way and the relation to PDP is also loose. As in the PDP the main point is the hierarchy of clustering concepts which is not exploited in this paper.  The experiments are insufficient because there are only MNIST and FMNIST tasks without any more realistic dataset such as commonly used CIFAR or TinyImgNet. The comparison with other baselines is insufficient either as mentioned above. And the memory size of memory-based reply methods should be equivalent to the memory cost of the VAE and GMM model, otherwise, it is not a fair comparison. The basic memory-based replay methods such as ER or Gdumb should be considered as a baseline as well.

Some minor issues:
1. Most of the citations should be in parentheses.
2. No enough explain of the GMM model, is it isotropic Gaussian? What's the definition of the covariate matrix?
3. Fig.1 is confusing as the pseudo points from the decoder should be fed to the classifier for replay, but the arrow is pointing to the encoder.
4. Fig.3 should be Tab.3



[1]. Rannen, Amal, et al. "Encoder based lifelong learning." Proceedings of the IEEE International Conference on Computer Vision. 2017.
[2]. Rao, Dushyant, et al. "Continual Unsupervised Representation Learning." Advances in Neural Information Processing Systems 32 (2019): 7647-7657.
[3]. Jeon, Ik Hwan, and Soo Young Shin. "Continual Representation Learning for Images with Variational Continual Auto-Encoder." ICAART (2). 2019.
[4]. van de Ven, Gido M., Hava T. Siegelmann, and Andreas S. Tolias. "Brain-inspired replay for continual learning with artificial neural networks." Nature communications 11.1 (2020): 1-14.
[5]. Pellegrini, Lorenzo, et al. "Latent replay for real-time continual learning." 2020 IEEE/RSJ International Conference on Intelligent Robots and Systems (IROS). IEEE, 2020
[6]. Silver, Daniel L., and Sazia Mahfuz. "Generating accurate pseudo examples for continual learning." Proceedings of the IEEE/CVF Conference on Computer Vision and Pattern Recognition Workshops. 2020.

**Summary Of The Paper:**

This paper proposes training a VAE in parallel with a classifier in continual learning and learning a Gaussian Mixture Model in the embedding space for sampling pseudo data points in experience replay.  The components in the Gaussian Mixture Model correspond to the classes of the data.

**Summary Of The Review:**

The contribution of this work is rather incremental and there is not enough discussion with related work, the experiments are not convincing either.

---

### Official Review · Reviewer_kyrh · 2021-11-08

**Correctness:** 3
**Technical Novelty And Significance:** 3
**Empirical Novelty And Significance:** 2
**Recommendation:** 5
**Confidence:** 4

**Main Review:**

This is a neatly conceptualized, well-motivated approach that is grounded in cognitive science theory. On one hand, in contrast to models with an explicit memory buffer, ICLA avoids a blow-up in memory complexity with an increasing number of samples/classes encountered. On the other hand, in contrast to models with synaptic constraints, ICLA doesn't restrict the expressivity of models. This also seems to be among initial efforts to jointly tackle the problem of continual and incremental learning.

However, several aspects of this manuscript need to be improved before it is publication-ready.

(i) Strength of empirical evaluation:
(a) Recent papers on continual learning evaluate their method on several benchmarks, including mini-ImageNet, CIFAR-100, CIFAR-10, etc. In comparison, the experiments presented in this paper seem too narrow in scope. It also does not provide an intuition for this algorithm's ability to scale.

(b) Is there a reason why these popular methods weren't included as part of the benchmark? Happy to stand corrected if the authors have a valid explanation.
Synaptic Intelligence [Continual Learning Through Synaptic Intelligence. Zenke et al (2017) ICML]
Elastic Weight Consolidation [Overcoming catastrophic forgetting in neural networks. Kirkpatrick et al (2017) PNAS]

(c) The improvement over competing methods for the 2T MNIST task is marginal. It is also unclear to me why Table 3 is not complete.

(ii) Literature review:
This method is also very related to "Online Class-Incremental Continual Learning with Adversarial Shapley Value" [Shim et al (2021) AAAI], where they introduce a scoring method to preserve latent decision boundaries. The authors should clarify similarities/differences.

(iii) Paper clarity:
This paper gives the impression that it was hurriedly put together. There are several grammatical and typographical errors, some of which I've tried to list below. A thorough proofread will enhance the quality of this manuscript.

Pg. 2 conceptsincrementally

Pg. 2 Incremental learning::

Pg. 2 incremental learningstems

Pg. 4 In Figure 1 caption: Based viewed enlarged

Pg. 4 "reduces the following" --> reduces to the following

Eq 2 (third term) there is a parenthesis mismatch

Pg. 7 "Figure 2a and present the learning curves...". Incomplete phrase

Pg. 7 "two tasks are define"

Table 3 is actually Figure 3 in the text.

Moreover, the citation style the authors use makes for a very unpleasant reading experience. In several places, the references directly interfere with the text. This needs to be fixed.

(iv) Other clarifications on the method:
(a) It is true that with an increasing number of samples/classes ICLA doesn't suffer from a memory bottleneck. However, the same cannot be said about the size of the "pseudo-dataset". Can the authors quantify how this will scale? This could prove to be a computational bottleneck on larger-scale tasks.

(b) "However, we can conclude that ICLA has a superior performance when the network size is small and using a memory buffer is not possible" --> Model parameter complexity results are not presented in this study. How did the authors arrive at this conclusion?

(c) $\gamma$ and $\lambda$ seem to be important hyperparameters of this model (for both learning and stability). Were these fixed apriori or dynamically adapted? The same goes for other hyperparameters such as the $N_{old}^{t}$. These choices need to be explained better.

(d) "Existence of this term indicates that our algorithm requires that internal distribution can be fit with a GMM distribution with high accuracy and this limits applicability of our algorithm. Note however, all parametric learning algorithms face this limitation". Have the authors considered directly modeling the density p(\phi(x)) using something like a flow network? This can circumvent the hacky GMM fitting.

(e) "This is because FMNIST data points are more diverse. As a result, generating pseudo-data points that look more similar to the original data points is easier for the MNIST dataset given that we are using the same network structure for both tasks". The way I understood from the paper, only the modes of the latent embedding density matter to "reinforce" knowledge from the past. Does this mean that the method is going to have bigger issues when moving to, say, mini-ImageNet?

(f) Can the authors give a sense of how much "drift" can be accommodated from time to time? For instance, would a full permutation of labels be amenable? To clarify, say, at time t+1 we change the labels of all 0s to 1s, all 1s to 2s, etc. There are no new classes added, but the mapping has been fully modified. I am just trying to get some intuition for what would happen in this case.

**Summary Of The Paper:**

In this paper, the authors propose ICLA, an approach to tackle the problem of incremental and continual learning. By keeping track of a model's "internal" representations of data, they identify and overcome the "drift" issue (as in continual learning). Particularly, they adopt an encoder-decoder style architecture to learn an embedding space that serves two purposes. (i) The embeddings from the encoder serve as an "internal" representation of data whose distribution is explicitly estimated using a Gaussian Mixture Model. (ii) By sampling embeddings from this GMM and passing them through the decoder, they turn this into a generative framework to support memory replay to overcome forgetting. When new classes are incorporated (as in incremental learning), additional components are added to the parameterization of the GMM. The authors provide some theoretical guarantees for error bounds and conduct empirical evaluations on MNIST and FMNIST.

**Summary Of The Review:**

This is a neat and well-thought-out idea. However, as I've expressed in the main review, some of the claims need further justification and the paper clarity needs to be significantly improved. I am willing to update my score if the authors are able to provide a convincing response!

---

### Decision · Program_Chairs · 2022-01-20

**Decision:**

Reject

**Comment:**

This paper tackles the challenge of continual learning. It approaches the problem by combining a Gaussian Mixture Model (GMM) to model concepts in a latent space and and a decoder system to generate new data points for pseudo-rehearsal and maintenance of previous information. When new concepts arrive, the GMM can be updated with rehearsal serving to prevent forgetting. The authors show competitive results on incremental learning of MNIST and FMNIST.

The scores were mostly below threshold, with one above threshold (5,3,5,6). The reviewers generally agreed the approach was interesting and they appreciated the theoretical treatments. However, there were a number of concerns, the central ones being the lack of clarity and the lack of convincing empirical demonstrations of scalability. The authors attempted to address the concerns, but they were not able to show good performance on larger datasets. The suggested this was due to the complexity of the encoding model, but they were unable to demonstrate this concretely. The reviewers' scores did not change, though, and the consensus was that this paper was not quite ready for publication. Given these considerations, and an average final score of 4.75, a decision of reject was reached.